# Evaluation of Glass-Ionomer versus Bulk-Fill Resin Composite: A Two-Year Randomized Clinical Study

**DOI:** 10.3390/ma15207271

**Published:** 2022-10-18

**Authors:** İlhan Uzel, Arzu Aykut-Yetkiner, Nazan Ersin, Fahinur Ertuğrul, Elif Atila, Mutlu Özcan

**Affiliations:** 1Department of Pediatric Dentistry, Ege University, Izmir 35040, Turkey; 2Independent Researcher, Izmir 35100, Turkey; 3Division of Dental Biomaterials, Clinic of Reconstructive Dentistry, Center of Dental Medicine, University of Zurich, 8032 Zurich, Switzerland

**Keywords:** glass-ionomer cement, bulk-fill composite, clinical performance, surface coat

## Abstract

Background: The aim of this split-mouth design research was to compare the clinical performance of a glass-ionomer cement system on Class I/II cavities against the clinical performance of bulk-fill resin composite restoration materials. Methods: Thirty-five patients were randomized and enrolled in the study, aged between 10 and 12 years, all of whom had a matched pair of permanent mandibular carious molars with similar Class I/II. A total of 70 restoration placements were performed. The patients were each given two restorations consisting of either a glass-ionomer cement with a nano-filled coating or a bulk-fill resin composite after the use of a self-etch adhesive. The cumulative survival rates were estimated using log-rank test and the Kaplan–Meier method. For comparison of the restorative materials in line with the modified Ryge, the McNemar test and the Wilcoxon signed rank test were employed. Results: With regard to retention, the glass-ionomer cement system and bulk-fill resin composite performed similarly in permanent molars in Class I/II cavities over a period of up to 24-months (*p* > 0.05). Over the 24-month period, Class I restorations showed statistically better survival rates than Class II restorations (*p* < 0.05). In the case of glass-ionomer cement systems, over the two-year period, more common chipping and surface degradations were observed. Conclusions: The glass-ionomer cement system and bulk-fill resin composite restorative materials display good clinical performance over a period of 24-months.

## 1. Introduction

A range of different materials are available for use in the restoration of permanent and primary teeth, such as conventional glass ionomer cement (GIC), compomer, resin-modified glass ionomer cement, and composite resins [1,2,3]. All of these have been commonly used in Europe as alternatives to amalgam [4]. The most-favored material generally is GIC, which, through the release of fluoride, offers extra protection against caries [1,3,4]. It should be noted, however, when a subsequent systematic review of clinical studies was carried out, the results showed no conclusive evidence of a caries-inhibitory effect [5]. Moreover, questions were raised by many studies as to whether conventional GIC does contain antibacterial properties, particularly regarding the viability of residual bacteria in carious dentin after all types of glass-ionomer restorations [4,5,6]. Other concerns that have been raised about GIC are a suspected tendency for microleakage and some deficiencies in physical properties, resulting in greater wear in stress-bearing cavities [7]. Another alternative material to GIC used in pedodontics clinics are types of composites. The new composite materials introduced to the market are described as ‘bulk-fill resin composites’ (BRC). The introduction of bulk-fill resin materials brought changes in their composition [8]. For this reason, regarding the application of traditional light-curing resin-based composites, it has been recommended as a gold standard that progressive increments of controlled thickness are used [9]. The maximum increment of 2 mm thickness has been generally accepted. Despite this, this is a time-consuming process in the case of deep cavities and in the case of contamination between the increments, and carries an increased risk of air bubble development [8]. Such factors as the resin shade/type filler amount and the intensity and spectrum of the activation light were all found to have a bearing on the cure depth of the resin composite [9,10]. These need to be accompanied by qualities of good adaptation and low residual stress, as required for all materials and filling techniques, and are essential through polymerization [10]. In recent years, dental practitioners have benefited from a significant expansion in the range and number of restorative materials available for use. A high-viscosity GIC system (GICs), developed over the years, has emerged as a prominent player in recent years due to its physical and chemical attributes, including fluorine release, adhesion to tooth tissue, biocompatibility, and its similarity to dentin tissue in terms of the thermal expansion coefficient [11]. The key features required in restorative materials are their similarity to dental tissue in terms of physical and chemical structure, high aesthetic and mechanical properties of restorative materials, and the ability to remain in the mouth for a long time while enabling the tooth to function as expected. In addition to these issues in child dentistry, it is essential that restorative materials can be applied to cavities quickly and easily. It is for this reason that GIC has emerged as a leading restorative material in pedodontics [12]. However, GIC does have some disadvantages when compared to resin composites [11,13]. These include the aesthetic appearance, lower fracture resistance, and greater occlusal abrasion. High-viscosity GIC, on the other hand, has improved the physical properties of the conventional product by optimizing the distribution of polyacids and particles. Recently, a GIC was launched using a cavity conditioner and light-curing nano-filler self-adhesive protective resin surface coating (G-Coat Plus) [13]. The availability of an increasing range of materials for use in restorative treatment has led to a growth in adhesive dentistry, and the use of composite resin with adhesive systems has become widespread [14]. It is recognized that the placement of the composite restoration on molar teeth can be a time-consuming process. It is recommended that the composite is placed in the cavity using the incremental technique. Bulk-fill-type composite materials have been developed to facilitate material placement and polymerization [9,10]. The term “bulk technique” is used because the chemically polymerized composites are placed into the cavity in a single layer. The name of the composite is also the name of the application technique [9]. It is known that the application of only a single layer reduces the clinical time for the physician and improves comfort for the patient [14]. The system which initiates polymerization in these composites also shortens the curing time. BRC have been reported to polymerize slowly due to modified methacrylate resins [15]. Kim et al. used four types of BRC material in their assessment, to test for micro-rigidity of the resin thickness. They concluded that BRC with a thickness application of 4 mm is appropriate [16].

The aim of our study was to compare the two-year clinical performance of GICs when used with self-adhesive resin, against BRC, used in conjunction with a self-etch adhesive.

## 2. Materials and Methods

The study protocol of this randomized, controlled, clinical trial was approved by the Ethics Committee of the Ege University (ethical code: 13-4/7). The aim of the study, its procedures, and its related risks were explained to the children and their parents before its commencement. Informed consent forms were signed by all the parents. A total of thirty children were included, attending the pediatric dentistry clinic of the Ege University. Twenty-five of the patients received two restorations while five of them received four restorations.

### 2.1. Patient Selection

The selection of the children was based on the following inclusion/exclusion criteria:

Inclusion criteria were as follows:(i)Children aged between 10 and 12 years who were healthy, without any known history of systemic illness;(ii)Children whose had a matched pair of maxillary and mandibular first/second permanent with an approximal or occlusal carious lesion of the same size;(iii)Children whose permanent molars consisted of a minimum 2.5 mm depth dentinal lesion;(iv)Children who had not received fissure sealant application;(v)Children who could attend the clinic regularly for controls throughout the 24 months.

Exclusion criteria were as follows:(i)Children with special needs;(ii)Uncooperative children;(iii)Children with molar incisor hypomineralization, enamel hypoplasia, or dental fluorosis;(iv)Children with parafunctional habits or bruxism;(v)Children with teeth that received excess or no load due to malocclusion.

### 2.2. Restoration Process

The operators all carried out their examinations under light using a mirror and probe, and all reported same-sized caries. We used a split-mouth design for the two different restorative materials. In the preparation of the cavities after removal of the caries, diamond rounds were used for the aerator and the drills. Tactile and normal optical criteria were used to verify the thorough removal of caries, after cavity preparation. Cavity sizes (depth) were measured with the help of the root canal instruments and endometer. The BRC was applied per layers, each approximately 4 mm thick. However, in the GICs group, it was placed in a single layer, regardless of the cavity depth. The power analysis of this study was performed based on the two different material evaluation periods to calculate the sample size. The sample size required for each group was determined to be at least 35 teeth (n = 70 total restoration) using G- power software version 3.1.9.7 for Windows (Heinrich Heine, Universität, Dusseldorf, Germany), for a power of 94%, the effect size of 1.73. In total, two operators performed 70 restoration placements.

One of the two materials was EQUIA, an encapsulated GICs marketed for use in all types of restorations. The other material used was a Tetric EvoCeram, applied in conjunction with a one-component bonding system AdheSe One F. The restorative material to be placed was randomly selected by a different dentist than the operator (double-blinded) before mixing the materials, and the treated dentin, in line with manufacturers’ instructions. The clinician used a SuperMat Universal matrix tensioning system (Kerr, Swiss) to maintain tight adaptation of the restorations.

Application of GICs: The cavities were prepared by a cavity regulator (GC Cavity Conditioner) for 10 s, with the help of cotton pellets. The cavities were washed with water and air-dried. After the glass-ionomer capsule was activated, the mixing device was mixed with Silver Mix 90 for 10 s and GIC material was placed in the cavity. Minimum moisture contamination was avoided. At the end of this period, G-Coat Plus was applied after superfluous diamond drillings and irradiated for 20 s. With the restorations set initially, the dentist carried out the necessary occlusal adjustment before applying G-Coat Plus over them.

Application of BRC material: AdheSE was applied to a cavity using brushed agitation with the aid of a VivaPen for 20 s. Adhesive material was polymerized for 10 s (Woodpecker, Yixing, China). BRC material was placed into the cavity to a maximum thickness of 4 mm, and then polymerized for 15 s. Silicon-based polishing material OptraPol was used for finishing the restoration after occlusion control. Any moisture that accumulated in this time was absorbed using cotton wool rolls. The clinician applied a dentin liner, Theracal in the deep dentinal lesions for pulpal protection, but excluded teeth with pulpal exposure. A matrix was used for all Class II restorations to maintain tight adaptation (Table 1).

### 2.3. Assesment and Statistical Analyses

The examiners, who were not involved in the placement procedures, evaluated all the restorations at 6, 12, 18, and 24 months. Cohen’s kappa values for inter-examiner and intra-examiner reliability, obtained after repeated examinations of 10% of the study group, were 0.91 and 0.82, respectively. The examiners performed the examinations during regular visits, using the modified Ryge criteria to evaluate the restorations (Table 2). In assessing the restorations, the location and type of tooth were recorded by the examiners.

Two of the authors of the study entered the data into spreadsheets and analyzed them using statistical software (SPSS 13.0, SPSS, Chicago). Cumulative survival rates were estimated using the log-rank test and the Kaplan–Meier method. For comparison of the restorative materials in line with modified Ryge, the McNemar test and the Wilcoxon signed rank test were employed.

## 3. Results

The mean age of the children was found to 11.2 years (Table 3). The cavity sizes ranged from 2.5 to 6 mm (mean: 3.7 mm). The 6-month cumulative survival rate for the Class I/II restorations was found to be 100% for both the GICs and BRC restorations.

The 12-month survival rate for Class I/II restorations was also found to be 100% for the GICs and BRC restorations. One-year clinical observation results were examined, and the retention rate of restorations was found to be 100%. The survival rates for both materials were 100% for a mean observation period of 10.4 months. Two of the GICs restorations showed chipping, but none for the BRC group. No secondary caries were observed in any restoration. Marginal discoloration scores were similar for both materials and none of the scores of Charlie, though the marginal fit was impaired.

The examiners evaluated 29 patients (68 restorations in total) after 24 months of follow-up. The patient follow-up rate was 96.6%.

The 24-month survival rate for Class I/II restorations was also found to be 97.1%and %100 for the GICs and BRC restorations, respectively. At the end of the 24-month follow-up, it was found that retention, secondary caries, marginal discoloration variables had no significant effect on the survival rates of the restorations (*p* > 0.05). After 24-months, results indicated only three cases of chipping for the GICs in Class II cavities. For the same period, no failures were observed for BRC. Secondary caries were observed in the case of GICs fillings (1 Charlie, 1 Bravo) and one BRC (1 Bravo). Marginal discoloration scores were similar for the two materials (7 GICs, 7 BRC) but surface porosity (10 GICs (2 Charlie), 3 BRC), and marginal deterioration (10 GICs (1 Charlie), 2 BRC) were all more prevalent with GICs in the Class II cavities. No statistically significant difference was apparent between the groups for either material (*p* > 0.05). Postoperative sensitivity scores were acceptable and similar for both materials (1 GICs and 1 BRC). Furthermore, no statistically significant difference was found after 24 months between the total numbers of Class I restorations patients that survived in the GICs and the BRC groups (*p* > 0.05).

Over the 24-month period, Class I restoration patients showed statistically better survival rates than Class II restoration patients. The cumulative survival percentages at 6, 12, and 24 months were calculated for each of the two restorative materials, according to class. This gave the distribution of the retained restorations according to the modified Ryge categories. However, no statistically significant difference was demonstrated between the two materials for any of the criteria, or with respect to the BRC or the GICs restorations (Table 4).

## 4. Discussion

Regarding new GICs, long-term clinical trials have assessed the GICs’ clinical performance in vivo studies in permanent molar teeth. Although some clinical studies have been carried out on the use of GIC as a restorative material in permanent teeth, only limited information is available on its use in relation to Class II cavities. We sought to establish whether reports were correct that the restorative GICs can be used effectively in both Class I and II cavities. In accordance with the modified Ryge criteria, to ensure as far as possible that objectivity was maintained, the two experienced experts were nominated, independent from the investigation team, to evaluate all the restorations. Our study therefore conformed with the recommendations of Hickel et al., who maintained that evaluation should be carried out by a researcher other than the physician who is applying the restorations [17]. A further check of the restorations was included at two years to complete the long-term study. In most clinical follow-up studies, modified Ryge evaluations criteria are used to assess the clinical performance of restorations [13,18]. Fridley et al. studied the clinical performance of GICs for the restoration of permanent molar teeth over 24 months. Their examination of 26 Class I and 125 Class II restorations in a total of 43 patients led to the conclusion that GICs was a suitable restoration material for permanent molar teeth. They concluded therefore that GICs can be used in Class I cavities of all sizes in permanent teeth, but it is more suitable for use in smaller Class II cavities [19]. In their study, Türkün and Kanık used GICs, GIC and Riva SC with two dissimilar surface coating materials (Fuji Varnish, G-Coat Plus). These were evaluated according to the modified Ryge criteria at the 6th, 12th and 18th month and the 6th year. They reported that, despite minor repairable defects, the overall clinical performance of GIC was found to be excellent. Compared to Riva SC, this was evident even in the case of large posterior Class II restorations at the final period of 6 years [20]. Khandewal et al. reported that the GICs was found to be successful at 88.8% for Class I cavities at the end of the 24-month follow-up [19]. Gürgan et al. carried out a four-year evaluation to determine the clinical performance of Gradia Direct restorations with GICs. No significant changes were reported with regard to color harmony, surface roughness, postoperative sensitivity, secondary caries, and anatomic form [13]. These accord with the findings of our study. Ersin et al. used composites and high-viscosity GIC in their study. They found that Surefil and Fuji IX composite materials with 419 Class I and Class II tooth restoration performed well. Their studies showed both restorative materials to be successful and there was no difference (*p* > 0.0.5), although the Class II restorations were found to be significantly more unsuccessful than the Class I restorations [21].

The criterion of retention is a crucial parameter in evaluating the clinical performance of restorations. The ADA reported that the retention rate should be at least 90% after18 months in order to be considered clinically successful [22]. Gürgan et al. reported a 24-month follow-up of Gradia Direct restorations with GICs applied to the surface. At the end of a four-year period, two restorations using glass-ionomer application were lost and the success rate was found to be 97.1% (100% in Class I restorations and 92.39% in Class II restorations) [13]. Hickel et al. reported that the loss rate of GIC ranged from 0 to 25.8% at the end of one year. It has been reported that the main cause of the losses in Class II restorations, as opposed to Class I, is high-pressure-related fractures [23]. In our study, both restoration materials were evaluated as successful according to ADA criteria for the 6-month and 24-month periods (100%, 97.1%). In our study, when the color matching of restorations was evaluated, initially, all restorations received the alpha score. Both restoration materials showed similar results at the end of 24 months and were deemed to be successful. Gürgan et al. came to analogous conclusions in their studies evaluating the clinical performance of the composite resin Gradia Direct Posterior with GICs. They reported that the color matching for both restoration materials was successful and remained similar after four years. Gürgan et al., though their results were not statistically significant, reported that Gradia Direct Posterior restorations of coloration compared well to GICs’s use of self-etch adhesive, with the formation of weaker adhesion on the edges of the cavity [13]. Loss of edge harmony between the tooth tissue and the material, interruption of the connection between the interface, application-related errors, and polymerization shrinkage can all play a role. This situation leads to microleakage, marginal coloration, and secondary caries [8,10]. Mahmoud et al. reported that coloration occurring between untreated enamel surfaces and the residual composite surplus could also be a factor [24]. In our study, there were no unsuccessful restorations in the 24-month evaluation period regarding the margin criterion. It is acknowledged that roughness on the surface of restorations is often due to insufficient finishing and polishing. This affects the overall clinical performance of restorations due to coloration, plaque accumulation, gingival irritation, and an increased susceptibility to recurrent bruising. Polishing tires, interface sanders, and blasters are used, though Sof-Lex discs have been reported to be more successful than other polishing systems [25]. Our study indicates that effective treatment using these procedures significantly affects the clinical performance of restorations. It is thought that the G-Coat Plus surface-coating material applied on GIC with the GICs also contributes to the success of restorations, in respect to the surface roughness criterion, after 24 months. In our study, when restorations were evaluated according to this criterion, it was observed that there were only two restorative C-D scores in the GICs group. In the clinical follow-up to our study no significant difference was found between the two different restoration materials. Postoperative sensitivity can stem from a range of different causes including trauma, excessive drying and marginal leakage during the procedure. It is thought that the coating and adhesive application of the Theracal LC (Bisco, Schaumburg, IL, USA) pulp coating, applied to deep cavities in GICs and BRC restorations, can be effective in decreasing sensitivity. We set out in this study to determine whether Class I/ II restorations using BRC would produce a higher retention rate in permanent teeth when compared with those carried out using GICs. Previously, clinicians favored resin-based composites for the treatment of permanent molars. Moreover, one of the suggested that resin-based composites, together with the adhesive systems recommended for use with them, might offer a longer survival time than GICs when used for Class II restorations [26]. Research by Hickel et al. found that annual failure rates for stress-bearing cavities of primary molars were 0 to 25.8 percent for GIC restorations, compared to 0 to 15 percent for resin-based composite restorations [17]. Our own findings showed the BRC to be satisfactory for Class I and II restorations in permanent teeth at the 24-month recall examination. In the case of the Class II restorations, the survival rate for the resin-based composite was shown to be higher than that for the GICs, though this difference was not statistically significant. It is possible, however, that a statistically significant difference might be obtained if the evaluation period was further extended. It also has the added benefit of reducing sensitivity when applied in deep cavities. The main failure characteristic for Class I/II restorations in the case of both materials was the loss of the restoration. We rarely observed caries at the margin, and secondary caries, which are generally a major cause of restoration failure, were also rarely in evidence. For both materials in our study, the surface texture, integrity, anatomical form, and marginal discoloration of all restorations placed were found to be satisfactory.

The GICs and BRC in both cavity types exhibited highly similar clinical performance over the observation period of two years. Consequently, the null hypothesis formulated at the beginning of this study was accepted. The GICs’ features are an important matter for clinical practice in pediatric dentistry. This is probably due to the chemical adhesion of the glass-ionomers system to the tooth structures surface without the need for adhesive systems, easy handling, and uncomplicated application completion [21]. In our study, the use of a GICs required relatively less time than BRC restorations. Moreover, regarding the GICs, their use in pediatric dentistry is perhaps preferred over BRC materials owing to their biocompatibility and, especially, the fluoride recharge/release from the glass-ionomer structure [5,12]. Continued development of the mechanical and physical properties of the both restorative materials will enable the construction of longer-lasting restorations. We believe that our study will provide useful guidance for future in-vivo and in-vitro studies on this subject and contribute to the further development of this area in the science of pedodontics.

## 5. Conclusions

Within the limitations of this study, we found that both GICs and BRC composite restorative materials exhibited satisfactory results after 2 years of clinical service. No statistically significant difference was found between the materials. However, Class I restorations showed statistically better clinical performance than Class II restorations according to the modified Ryge criteria for GICs. Based on these results, both the BRC and GICs can be recommended for use by clinicians as alternative options for Class I/II restorations and for deep cavities. However, it will be necessary in the longer term to continue to monitor the survival rate of GICs Class II restorations because of the failed Class II restorations in our study.

## Figures and Tables

**Table 1 materials-15-07271-t001:** Description of materials used in this study.

Material (Lot No)	Type	Manufacturer	Composition
**EQUIA** **(Lot No: 1005265)**	High-viscosity encapsulated glass-ionomer cement system	GC, Leuven, Belgium	Powder: 95% strontium fluoro alumino-silicate glass, 5% polyacrylic acid. Liquid: 40% aqueous polyacrylic acid
**EQUIA Coat** **(Lot No: 0811130)**	Low-viscosity nano filled surface coating resin	GC, Tokyo, Japan	40–50% methyl methacrylate, 10–15% colloidal silica, 0.09% camphorquinone, 30–40% urethane methacrylate, 1–5% phosphoric ester monomer
**GC Cavity conditioner** **(Lot No:1102151)**	Cavity conditioner	GC, Tokyo, Japan	77% distilled water, 20% polyacrylic acid, 3% aluminum chloride hydrat
**Tetric EvoCeram** **(Lot No: P63356)**	Bulk-fill composite	Ivoclar Vivadent, Schaan, Liechtenstein	Matrix: 19% weight, Bis-GMA, UDMA, Bis-EMA Filler: %81 weight, Ba-Al-Si glass, prepolymer (including 17% filler (monomer, glass filler, prepolymers) and ytterbium fluoride), spherical mixed oxide
**AdheSe One F** **(VivaPen)** **(Lot No:L17747)**	Self-etching, one component dentin bonding system	Ivoclar Vivadent, Schaan, Liechtenstein	Derivatives of bis-acrylamide, water, alcohol, bis-methacrylamidehydrogen phosphate, amino acid acrylamide, hydroxyl alkyl methacrylamid, alkyl sulfonic acid acrylamide, highly dispersed silicon dioxide, catalysts and stabilizers, potassium fluoride
**OptraPol** **(Lot No:UL0838)**	Silicone based composite polishing material	Ivoclar Vivadent, Schaan, Liechtenstein	Silicone rubber and diamond particles, aluminium oxide, iron oxide and Irgazin
**Theracal LC** **(Lot No: 1400007511)**	Light-cured, resin-modified calcium silicate filled liner	Bisco, Illinois, USA	Calcium silicate, Portland cement, PEG-DMA

**Table 2 materials-15-07271-t002:** Modified Ryge evaluation criteria.

Criteria	Scores	Definition
Color Match	Alpha Bravo Charlie	The restoration matches adjacent tooth structure in color, shade and transluceny Mismatch in within an acceptable range of color, shade and translucency This mismatch is outside acceptable range of tooth color and translucency
Marginal Discoloration	Alpha Bravo Charlie	Absence of marginal discoloration between the restoration Presence of marginal discoloration slightly Evident marginal discoloration penetrated toward the pulp direction
Marginal Integrity	Alpha Bravo Charlie	No visible evidence of crevice Visible crevice, explorer will penetrate Crevice in which dentin or the base is exposed
Surface Texture	Alpha Bravo Charlie	Smooth surface Slightly rough or pitted Rough, cannot be refinished, fracture on the surface of the restoration
Retention	Alpha Bravo Charlie	Complete retention of the restoration Mobilization of the restoration, still present Loss of the restoration
Secondary Caries	Alpha Bravo Charlie	No evidence of caries Evident of caries along the margin of the restoration Restoration is replaced because of caries
Postoperative Sensitivity	Alpha Bravo	No post-operative hypersensitivity Experience of dentinal hypersensitivity

**Table 3 materials-15-07271-t003:** Demographic characteristics of patients with GICs and BRC groups.

	GICs N (%)	BRC N (%)
* **Gender** *		
Male	8 (47.05%)	9 (52.95%)
Female	7 (53.8%)	6 (46.2%)
* **Age** *		
10	4 (57.1%)	3 (42.9%)
11	4 (44.4%)	5 (55.6%)
12	7 (50%)	7 (50%)

**Table 4 materials-15-07271-t004:** The modified Ryge scores of restorations made with GICs and BRC applied in the study after treatment (post-op.), 6-month, 12-month, 18-month, and 24-month follow-ups.

Modified Ryge Criteria	Study Materials									
	GICs (EQUIA)					BRC (Tetric EvoCeram)				
Evaluation Periods	Post-Op.	6th Month	12 Month	18th Month	24th Month	Post-Op.	6th Month	12 Month	18th Month	24th Month
Color Match	(A = 35)	(A = 35)	(A = 34) (B = 1)	(A = 34) (B = 1)	(A = 32) (B = 2)	(A = 35)	(A = 35)	(A = 35)	(A = 35)	(A = 34)
Marginal Discoloration	(A = 35)	(A = 34) (B = 1)	(A = 34) (B = 1)	(A = 31) (B = 4)	(A = 27) (B = 7)	(A = 35)	(A = 35)	(A = 34) (B = 1)	(A = 33) (B = 2)	(A = 27) (B = 7)
Marginal Integrity	(A = 35)	(A = 35)	(A = 31) (B = 4)	(A = 27) (B = 8)	(A = 24) (B = 9) C = 1)	(A = 35)	(A = 35)	(A = 33) (B = 2)	(A = 33) (B = 2)	(A = 32) (B = 2)
Surface Texture	(A = 35)	(A = 35)	(A = 33) (B = 2)	(A = 30) (B = 5)	(A = 24) (B = 8) (C = 2)	(A = 35)	(A = 35)	(A = 35)	(A = 34) (B = 1)	(A = 31) (B = 3)
Retention	(A = 35)	(A = 35)	(A = 33) (B = 2)	(A = 33) (B = 2)	(A = 31) (B = 2) (C = 1)	(A = 35)	(A = 35)	(A = 35)	(A = 35)	(A = 34)
Secondary Caries	(A = 35)	(A = 35)	(A = 35)	(A = 34) (B = 1)	(A = 32) (B = 1) (C = 1)	(A = 35)	(A = 35)	(A = 35)	(A = 35)	(A = 33) (B = 1)
Postoperative Sensitivity	(A = 35)	(A = 35)	(A = 35)	(A = 34) (B = 1)	(A = 33) (B = 1)	(A = 35)	(A = 35)	(A = 34) (B = 1)	(A = 34) (B = 1)	(A = 33) (B = 1)

Case numbers and scores are given in parentheses. (A = Alpha, B = Bravo, C = Charlie).

## Data Availability

Derived data supporting the findings of this study are available from the corresponding author on request.

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
