# Peer review of "Evaluation of Glass-Ionomer versus Bulk-Fill Resin Composite: A Two-Year Randomized Clinical Study"

_materials, 2022, doi:10.3390/ma15207271_

Round 1

Reviewer 1 Report

Introduction is long. Please summarize the more relevant information.

Conclusions must be more directly presented, in a format shorter than the presented.

Author Response

Point 1: “Introduction is long. Please summarize the more relevant information.”

Response 1: More detailed information is abbreviated and summarized in the introduction section.

Point 2: “Conclusions must be more directly presented, in a format shorter than the presented.”

Response 2: The results were presented more directly in a shorter format, in line with the suggestions of other reviewers.

Reviewer 2 Report

In the abstract, “For comparison of the re-24 storative materials in line with Modified Ryge, the McNemar test and the Wilcoxon signed rank test 25 were employed (p= 0.05).”, the p value was given for no reason, it seem to be confused with significant level.

Lot number, expiration data of the materials or the period of the clinical experiment starting from which year. EQUIA was firstly launched in Nov. 2009. With years a few generations of material may be developed.

Material “Theracal” should be explained with manufacturer and composition.

It is not clear what it means for these sentences:

L154  “material to be placed was randomly (heads or teals)

L167 “Adhesive material was diluted with air

L169 “The finishing process used silicone-based polishing material”

L185 “Differences were significant at P= .05”

The texts in Table 2 should be better aligned.

In the result, a descriptive table should be included to show the basic demography information of the two groups and the parameters of the restoration. The depth of application per layer should be described as well. In the method section it says 30 participants with 70 restorations applied. So how the restorations were allocated in each group?

The p-values should be provided in order to compare with a significant level, and the statistical analysis method should be accompanied by the p-values, such as L-203, " no significant effect".

The data of marginal discoloration scores, cumulative survival percentages, and distribution of the retained restorations are missing in the results.

Author Response

Point 1: “In the abstract, “For comparison of the restorative materials in line with Modified Ryge, the McNemar test and the Wilcoxon signed rank test were employed (p= y the p value was given for no reason, it seem to be confused with significant level.”

Response 1: The p value was removed

Point 2: “Lot number, expiration data of the materials or the period of the clinical experiment starting from which year. EQUIA was firstly launched in Nov. 2009. With years a few generations of material may be developed.” 

Response 2: Lot numbers of each product were added. Also, the period of the clinical experiment starting from 2014 year.

Point 3: “Material “Theracal” should be explained with manufacturer and composition.” 

Response 3: The “Theracal” material is described in Table 1 with manufacturer and composition.

Point 4: “It is not clear what it means for these sentences:

             L154 “material to be placed was randomly (heads or teals)”

             L167 “Adhesive material was diluted with air”

             L169 “The finishing process used silicone-based polishing material”

             L185 “Differences were significant at P= .05””

Response 4: Sentences were corrected to be more understandable or irrelevant sentences were deleted.

Point 5: “The texts in Table 2 should be better aligned.”

Response 5: Text in Table 2 is arranged and checked for alignment.

Point 6: “In the result, a descriptive table should be included to show the basic demography information of the two groups and the parameters of the restoration. The depth of application per layer should be described as well. In the method section it says 30 participants with 70 restorations applied. So how the restorations were allocated in each group?”

Response 6: Descriptive table were included. A total of thirty children were included, twenty-five of them receiving two restorations and five of them receiving four restorations.

Point 7: “The p-values should be provided in order to compare with a significant level, and the statistical analysis method should be accompanied by the p-values, such as L-203, " no significant effect".”

Response 7:  Statistical analysis method accompanied by p-values ​​and p-values ​​provided for comparison with a significant level

Point 8: “The data of marginal discoloration scores, cumulative survival percentages, and distribution of the retained restorations are missing in the results.”

Response 8: Marginal coloration scores are added, missing their survival and disappearance results. The results had presented in a table.

Reviewer 3 Report

It is a topic that is very insufficient to draw the reader's interest as a mediocre research content. Moreover, the structure of the manuscript makes it very difficult for the reader to understand.

The authors must edit the composition of this manuscript in order to more effectively convey the experimental results to the readers.

Author Response

Point 1: “It is a topic that is very insufficient to draw the reader's interest as a mediocre research content. Moreover, the structure of the manuscript makes it very difficult for the reader to understand.”

Response 1: We are deeply sorry for the concerns of the reviewer. We sincerely appreciate the comment of the reviewer. According to the reviewer's suggestion, the old manuscript was replaced with revised documents.

Point 2: “The authors must edit the composition of this manuscript in order to more effectively convey the experimental results to the readers.”

Response 2: Extensive rechecking of grammar, and punctuation was achieved with the help of professional English language editing services. Requirements for improving the article and making it suitable for publication were made in line with the comments of the reviewers. We wanted to significantly improve the quality of the manuscript.

Reviewer 4 Report

Dear Authors

I have read with great interest and emotion your article “Evaluation of Glass-ionomer versus bulk-fill composite: a two years randomized clinical study”. In general, I consider the manuscript provides important results but it is not well written. A professional language review would help to order your ideas and make the manuscript more understandable. In addition, I have a few concerns and suggestions that must be attended.

More uniformity and order in the text are needed.

For example, Uniformity in naming materials.

Line 18: “encapsulated-high-viscosity glass-ionomer”

Line 22 and 26: “glass-ionomer cement”

Line 39: “glass ionomer cement (GIC)”, besides this is not the first time you mention it. “GIC” should be written from line 36 and from there always write “GIC” and never more the complete words.

Some examples of where it should write GIC are Lines: 68, 81, 86, 150, 159, 198, and 385. But you need to locate them all and unify the name.

In the same way, you should unify if you are going to call it “EQUIA” or “GIC”, because you use it interchangeably and it becomes confusing. Furthermore, if you are going to call it “EQUIA”, then you should call the other group “Tetric EvoCeram”.

The non-uniform use of terms is also used for “bulk-fill resin composite (BRC)”. On line 51, you mention how it will be abbreviated, but you continue to use the full words and non-uniforms in the text. For example in lines: 106, 108, 116, 117, 128, 132, 165, 171, 198, 206, 213, 225, 344, 364, 373, 385, and 387.

What is the difference between the following terms? in the document you use them interchangeably. You must unify them and use the most correct term for what you measured. “performance” line 17, “survival rates” line 23, 183, 192, 196 “clinical performance” line 130, “clinical success” line 238, “retention rate” line 196, “dropout rate” line 201

The introduction is very long (You have 100 lines of introduction!!) and includes topics that are not specifically relevant to the study. If the reader want to know so many details about materials, the reader would go to a book or a review article on the properties of materials.

Besides that, you have a lot of text without citations, for example

Line 61-65

Line 72-78

Line 81-90

Line 95-111

Line 117-126

All those texts need at least one citation, but it will be better to reduce the introduction to the essential minimum.

The introduction also needs to be reordered by presenting the ideas logically, in sequence and without repeating information. For example line 51 and line 122 say the same thing.

Line 50: ensures that ..”new composite materials introduced to the market…” however these have already been on the market for several years, you even mention works from a few years ago that have already been done with these materials. They are no longer "new"

Lines 130-134. Check the information in parentheses, in some cases within the same parentheses you write the name of the product, the brand and the origin, while in other cases you write double parentheses, one with the name and the other with the brand and origin. In addition, it is no longer necessary to repeat this information later in the text, it must be eliminated, for example, from line 149, 151, 153, 158, and others... it should not even be repeated in table 1 or leave this information in table1 and remove it from the text.

Materials and methods

It needs to be restructured; it is very difficult to read because it has jumbled information. Usually, first the patients are described, where they were treated, their selection criteria (you mention a selection criterion almost at the end in line 174), bioethical aspects and then it is described how the experiments were carried out. It is necessary to order this section, divide it into subsections and give more information on each of them, for example, where these children were treated? What information can be obtained about their clinical characteristics? It is also mentioned that informed consent was obtained? From the children? From their parents? And the assent?

Line 150 what does (GICs) mean?, is this another abbreviation?

Line 161. What does “CIS” mean?

Line 180. What does “USPHS” mean? and why is it not used uniformly in the text?

Tables need uniformity and revision too. For example, you write “Tokyo, Japan” in parentheses, but not the others.

The numbers 79-81 and 60-61 are percentages?

“Silicone rubber” must be capitalized.

The “retention” criterion repeats its definition.

Results

The results start by saying that: …”cavity sizes ranged from 2.5 to 6mm..” but it is not clear where it was said that the cavities were measured or how they were measured.

Line 203. ..”it was found that these variables…” which variables exactly?

In general, the results are difficult to understand, you should try to present them in an orderly way in a table.

Discussion, the same happens as with the introduction, you have almost 150 lines of discussion, much of what was put in it has nothing to do directly with discussing your results, you even cite other works without it being clear why you cite them, for example, lines 318-323 are about postoperative sensitivity, did you measure sensitivity in your patients?

It is necessary to reduce the discussion by half, everything that is said there must have a meaning and serve to discuss your results.

In many cases citations are missing as in the introduction.

The discussion repeats information that has already been given, for example the information for lines 240-243 is in table 1.

The discussion contains information that is not consistent with what was said in the materials and methods, for example line 233 says…” a singles expert was nominated, independent from the investigation team, to evaluate all the restorations…” while in line 136, 175 and 179 you speak in the plural of the examiners and even mention that ..inter-examiner and intra-examiner reliability was obtained in line 177. So was it one or several examiners? this detracts from the credibility of the experiment.

Line 230.…”class I/II caries do not exist.

On the other hand, you need to rewrite the conclusion. It must be restructured, reduced and eliminate assertions that are not the result of your work, for example the last three of lines 383 -388. You cannot guarantee that glass-ionomers or the bulk-fill resin based composites can be used or that they have success because that generalizes to all glass-ionomers and bulk fill composites and you only studied two commercial brands and there are many more.

In the conclusion you mention that …” Class I restorations showed statistically better survival rates than Class II restorations…” but you do not show this information in the results. You should show the information separately for class I and class II in the results and then discuss it in the discussion.

Why in lines 375-376 do you only mention that class II must be monitored, and why the class I not?

The conclusion of the abstract should also be corrected. In it you conclude that …” two incidences of chipping were recorded and more common surface degradations were observed…” Do you really want to show that as the conclusion of the work? Remember that the abstract is the second approach of the readers to the article (the first is the title).

It is also necessary to review the references one by one, as they do not always have the same format. It must be unified, for example citations 15,21,22 among others.

The language review will allow you to avoid texts that are not understood and are confusing, such as:

Line 136. ..”The examiners all carried out their examinations……”

Line 222. …” the EQUIA is a long-term clinical trial of high viscosity…”

Among many others that make reading difficult.

I say it again, the work is very interesting, but the quality of the written document needs to be greatly improved.

Author Response

Point 1: “I have read with great interest and emotion your article “Evaluation of Glass-ionomer versus bulk-fill composite: a two years randomized clinical study”. In general, I consider the manuscript provides important results but it is not well written. A Professional language review would help to order your ideas and make the manuscript more understandable. In addition, I have a few concerns and suggestions that must be attended to.” 

Response 1: We are grateful to you for your kind interest to improve the scientific value of our paper; your comments are greatly appreciated. Additionally, our responses to each of the comments are stated below and in a separate file. Extensive recheck of the manuscript with the help of professional language editing services.

Point 2: “More uniformity and order in the text are needed.

For example, Uniformity in naming materials.

Line 18: “encapsulated-high-viscosity glass-ionomer”

Line 22 and 26: “glass-ionomer cement”

Line 39: “glass ionomer cement (GIC)”, besides this is not the first time you mention it. “GIC” should be written from line 36 and from there always write “GIC” and never more the complete words.

Some examples of where it should write GIC are Lines: 68, 81, 86, 150, 159, 198, and 385. But you need to locate them all and unify the name.

In the same way, you should unify if you are going to call it “EQUIA” or “GIC” because you use it interchangeably and it becomes confusing. Furthermore, if you are going to call it “EQUIA”, then you should call the other group “Tetric EvoCeram”.”

Response 2: Glass ionomer cement is written “GIC” everywhere and EQUIA is called “GICs”

Point 3: “The non-uniform use of terms is also used for “bulk-fill resin composite (BRC)”. On line 51, you mention how it will be abbreviated, but you continue to use the full words and non-uniforms in the text. For example in lines: 106, 108, 116, 117, 128, 132, 165, 171, 198, 206, 213, 225, 344, 364, 373, 385, and 387.”

Response 3: The uniform use of terms was also used for “bulk-fill resin composite”.

Point 4: “What is the difference between the following terms? in the document you use them interchangeably. You must unify them and use the most correct term for what you measured. “performance” line 17, “survival rates” line 23, 183, 192, 196 “clinical performance” line 130, “clinical success” line 238, “retention rate” line 196, “dropout rate” line 201”

Response 4: We used the most correct term for what we measured in the revised manuscript.

Point 5: “The introduction is very long (You have 100 lines of introduction!!) and includes topics that are not specifically relevant to the study. If the reader want to know so many details about materials, the reader would go to a book or a review article on the properties of materials.”

Response 5: More brief information is abbreviated and summarized in the introduction section.

Point 6: “Besides that, you have a lot of text without citations, for example

Line 61-65

Line 72-78

Line 81-90

Line 95-111

Line 117-126

All those texts need at least one citation, but it will be better to reduce the introduction to the essential minimum.”

Response 6: The introduction reduced to the essential minimum.”

Point 7: “The introduction also needs to be reordered by presenting the ideas logically, in sequence and without repeating information. For example line 51 and line 122 say the same thing.”

Response 7: The introduction by presenting the ideas logically, in sequence and without repeating information reordered.

Point 8: “Line 50: ensures that ..”new composite materials introduced to the market…” however these have already been on the market for several years, you even mention works from a few years ago that have already been done with these materials. They are no longer "new"”

Response 8: The word "new" was removed from the sentence.

Point 9: “Lines 130-134. Check the information in parentheses, in some cases within the same parentheses you write the name of the product, the brand and the origin, while in other cases you write double parentheses, one with the name and the other with the brand and origin. In addition, it is no longer necessary to repeat this information later in the text, it must be eliminated, for example, from line 149, 151, 153, 158, and others... it should not even be repeated in table 1 or leave this information in table1 and remove it from the text.”

Response 9: We save the information in table1 and remove it from the text.

Point 10: “It needs to be restructured; it is very difficult to read because it has jumbled information. Usually, first the patients are described, where they were treated, their selection criteria (you mention a selection criterion almost at the end in line 174), bioethical aspects and then it is described how the experiments were carried out. It is necessary to order this section, divide it into subsections and give more information on each of them, for example, where these children were treated? What information can be obtained about their clinical characteristics? It is also mentioned that informed consent was obtained? From the children? From their parents? And the assent?”

Response 10: The methodology has been arranged in a simple and easy-to-understand manner and changes have been made to the Tables according to your requests. Also, this section is divided into subsections and given more information.

Point 11: “Line 150 what does (GICs) mean?, is this another abbreviation?”

Response 11: “GICs” means glass ionomer cement system=EQUIA

Point 12: “Line 161. What does “CIS” mean?”

Response 12:  Changed to “GIC”

Point 13: “Line 180. What does “USPHS” mean? and why is it not used uniformly in the text?”

Response 13: Modified Ryge used uniformly in the text, “USPHS” removed

Point 14: “Tables need uniformity and revision too. For example, you write “Tokyo, Japan” in parentheses, but not the others.”

Response 14: The tables have been revised.

Point 15: “The numbers 79-81 and 60-61 are percentages?” 

Response 15: “81” weight percent, “61” volume percent. The tables have been revised

Point 16: “Silicone rubber” must be capitalized.”

Response 16: The tables have been revised.

Point 17: “The “retention” criterion repeats its definition.”

 Response 17: The tables have been revised.

Point 18: “The results start by saying that: …”cavity sizes ranged from 2.5 to 6mm..” but it is not clear where it was said that the cavities were measured or how they were measured.”

Response 18: Cavity sizes (depth) were measured with the help of the root canal instruments and endometer after cavity preparation.

Point 19: “Line 203. ..”it was found that these variables…” which variables exactly?”

Response 19: retention, secondary caries, marginal discoloration

Point 20: “In general, the results are difficult to understand, you should try to present them in an orderly way in a table.”

 Response 20: The results had presented in a table.

Point 21: “Discussion, the same happens as with the introduction, you have almost 150 lines of discussion, much of what was put in it has nothing to do directly with discussing your results, you even cite other works without it being clear why you cite them, for example, lines 318-323 are about postoperative sensitivity, did you measure sensitivity in your patients?”

Response 21: The discussion section was restructured, reduced, and eliminated assertions that are not the result of our work. Postoperative sensitivity of patients was measured.

Point 22: “It is necessary to reduce the discussion by half, everything that is said there must have a meaning and serve to discuss your results.”

Response 22: The discussion section was restructured, reduced, and eliminated assertions that are not the result of our work.

Point 23: “In many cases citations are missing as in the introduction.”

Response 23: Carefully reviewed and checked again.

Point 24: “The discussion repeats information that has already been given, for example the information for lines 240-243 is in table 1.”

Response 24: Lines 240-243 were removed.

Point 25: “The discussion contains information that is not consistent with what was said in the materials and methods, for example line 233 says…” a singles expert was nominated, independent from the investigation team, to evaluate all the restorations…” while in line 136, 175 and 179 you speak in the plural of the examiners and even mention that ..inter-examiner and intra-examiner reliability was obtained in line 177. So was it one or several examiners? this detracts from the credibility of the experiment.”

Response 25: Line 233…. changed to "two experienced experts were assigned to evaluate all restorations."

Point 26: “Line 230.…”class I/II caries do not exist.”

Response 26: The text “class I/II caries” has been removed and changed to occlusal/approximal caries.

Point 27: “On the other hand, you need to rewrite the conclusion. It must be restructured, reduced and eliminate assertions that are not the result of your work, for example the last three of lines 383 -388. You cannot guarantee that glass-ionomers or the bulk-fill resin based composites can be used or that they have success because that generalizes to all glass-ionomers and bulk fill composites and you only studied two commercial brands and there are many more.”

Response 27: The conclusion section was restructured, reduced, and eliminated assertions that are not the result of our work. We recommend that only which we use glass ionomers or bulk-fill resin-based composites can be used.

Point 28: “In the conclusion you mention that …” Class I restorations showed statistically better survival rates than Class II restorations…” but you do not show this information in the results. You should show the information separately for class I and class II in the results and then discuss it in the discussion.”

Response 28: Summary with information for Class I and II was shown in the results

Point 29: “Why in lines 375-376 do you only mention that class II must be monitored, and why the class I not?”

Response 29: Class II should be monitored more carefully; because all the low (failed) scores were seen in that group restorations. In addition, in the literature, it is recommended to attention to Class II restorations.

Point 30: “The conclusion of the abstract should also be corrected. In it you conclude that …” two incidences of chipping were recorded and more common surface degradations were observed…” Do you really want to show that as the conclusion of the work? Remember that the abstract is the second approach of the readers to the article (the first is the title).”

Response 30: Abstract edited.

Point 31: “It is also necessary to review the references one by one, as they do not always have the same format. It must be unified, for example citations 15,21,22 among others.”

Response 31: References were checked again.

Point 32: “The language review will allow you to avoid texts that are not understood and are confusing, such as:

Line 136. ..”The examiners all carried out their examinations……”

Line 222. …” the EQUIA is a long-term clinical trial of high viscosity…”

Among many others that make reading difficult.”

Response 32: Extensive rechecking of grammar, and punctuation has been achieved with the help of professional English language editing services.

Point 33: “I say it again, the work is very interesting, but the quality of the written document needs to be greatly improved.”

Response 33: We sincerely appreciate your comments. We wanted to significantly improve the quality of the manuscript

Round 2

Reviewer 2 Report

After revision, the manuscript quality has been significantly improved.

Author Response

Thank you very much for your comment.

Reviewer 3 Report

This revised manuscript has undergone many revisions and appears to be very interesting to read and understand by readers compared to the version originally submitted.

Author Response

Thank you very much for your comment and your valuable contribution.

Reviewer 4 Report

In the last review, I asked for uniformity in the text, it has been improved, but it still does not exist completely. For example:

In line, 93 the authors state that they will call the "high-viscosity GIC system EQUIA" as "GICs" and they do it well during the following mentions, until lines 187 and 188, where they even write again that "EQUIA is a high -viscosity GIC system and re-establish that it will be called “GICs”.

Later on line 199, they rewrite "glass ionomer". During several pages they do it correctly until in the discussion; they write “EQUIA” many times again (lines 310, 334, 335, 337, 340, and several more!)

In line 57 they mention "bulk-fill resin composites" for the first time and establish that it will be abbreviated as "BRC", however in lines 209 416, 431 they rewrite it in its complete form.

Lines 200, 204 “GC, Europe” or “G-Coat Plus “are different?

These are just a few examples of uniformity

In the last revision, I asked to review very critically the introduction. Although they modified it, it is still very long, and contains texts that have no logical sequence, contradictory texts, texts that repeat information, and texts without citations, for example:

Line 48-49: …” It should be noted, however, that these studies were mostly not carried out not in-vivo….”

What studies are they referring to? In the sentence before that, they are talking about the "GIC" and not about any study.

Line 56: the sentence ….”On the other hand, the composite materials introduced to the market, described as ‘bulk-fill resin composites’ (BRC), claimed to be curable for thicknesses of 4 mm….”

This sentence comes out of nowhere, the sentence before it talks about some deficiencies in physical properties of GIC. Why is the 4mm in the BCR suddenly relevant? This makes no sense.

Finally: lines 50 to 54 establish that the GIC has problems, limitations such as “no conclusive evidence of caries inhibitory effect” or “it suspected tendency for microleakage” or “deficiencies in physical properties”… but then in lines 75-77 (only 20 lines later) they say that … “GIC has emerged as a prominent player through its SUPERIOR PHYSICAL, chemical and mechanical ATTRIBUTES, including fluorine release, adhesion to tooth tissue, biocompatibility, and its similarity to dentin tissue in terms of thermal expansion coefficient….”

So they think GIC are good, bad, or what? This is very confusing.

In the last revision I also requested to add citations, they make specific statements without citations, two examples are:

Line 103: …”One of the drawbacks of composite materials polymerization, with 1-3% shrinkage during curing…”

Line 105: … Polymerization shrinkage causes post-restoration precision, failure of marginal adaptation and an increase in microleakage values…”

Line 182: what does the following sentence mean? “…The power analysis of this study was performed based on the two different materials…” What does this imply? What was the result of the analysis? what does it mean to write this?

Writing “the first” and “the second” in lines 187 and 189 confuses when later they write “randomly” in line 190

Line 193. Double-blinded?  A double blind means that neither the participants nor the investigator, in this case, the dentist knows which intervention is receiving. This is not possible because the dentist used two protocols for the placement of the different materials and clearly knows what he is placing.

The first sentence of the results (line 254) says…” The ages of the children ranged from 10 to 12 years”… but this is not necessary because that is precisely a selection criterion “ line 154

In discussion, the same happens as with the introduction, it is necessary to reduce the discussion, everything that is said there must have a meaning and serve to discuss the results. In many cases, citations are missing as in the introduction.

The first sentence of the discussion on lines 310-311 reads: …” Nowadays, EQUIA is a long-term clinical trials assessing GICs’ clinical performance in-vivo studies in permanent molar teeth are limited…”.

What does that mean?

What is the difference between the following sentences?

Lines 464-465: … the both GICs and  BRC composite restorative materials exhibited satisfactory results after 2 years of clinical service…..

Lines 473-474: … The GICs and BRC restorative materials display good clinical performance with a period of 24 months

Both are in the conclusion with only eight lines of separation

Finally, the following sentences are not a conclusion of the study; in any case, they could be part of the discussion:

475-476: …”Continued development of the mechanical and physical properties of the both restorative 475 materials will enable the construction of longer-lasting restorations…”

478-480: …”We believe that our study will provide useful guidance for future in-478 vivo and in-vitro studies on this subject and contribute to the further development of this 479 area in the science of pedodontics…”

I say it again, the work is very interesting, but the quality of the written document needs to be greatly improved.

Author Response

POINT 1: In the last review, I asked for uniformity in the text, it has been improved, but it still does not exist completely. For example:

In line, 93 the authors state that they will call the "high-viscosity GIC system EQUIA" as "GICs" and they do it well during the following mentions, until lines 187 and 188, where they even write again that "EQUIA is a high -viscosity GIC system and re-establish that it will be called “GICs”.

RESPONSE 1: The glass ionomer cement system is mentioned as GICs instead of the brand name EQUIA.

POINT 2: Later on line 199, they rewrite "glass ionomer". During several pages they do it correctly until in the discussion; they write “EQUIA” many times again (lines 310, 334, 335, 337, 340, and several more!)

RESPONSE 2: It has changed as GICs are also in the discussion section.

POINT 3: In line 57 they mention "bulk-fill resin composites" for the first time and establish that it will be abbreviated as "BRC", however in lines 209 416, 431 they rewrite it in its complete form.

RESPONSE 3: All expressions written as bulk-fill resin composites in complete form have been changed to the BRC abbreviation.

POINT 4: Lines 200, 204 “GC, Europe” or “G-Coat Plus “are different?

 RESPONSE 4: Lines 200, 204 “GC, Europe” or “G-Coat Plus” are different expressions. Also, the phrase "GC, Europe" has been removed to avoid confusion.

POINT 5: These are just a few examples of uniformity

RESPONSE 5: The asked changes were done to provide uniformity in the article.

POINT 6: In the last revision, I asked to review very critically the introduction. Although they modified it, it is still very long, and contains texts that have no logical sequence, contradictory texts, texts that repeat information, and texts without citations, for example:

Line 48-49: …” It should be noted, however, that these studies were mostly not carried out not in-vivo….”

What studies are they referring to? In the sentence before that, they are talking about the "GIC" and not about any study.

RESPONSE 6: The word that needs to be removed in lines 48-49 have been removed.

POINT 7: Line 56: the sentence ….”On the other hand, the composite materials introduced to the market, described as ‘bulk-fill resin composites’ (BRC), claimed to be curable for thicknesses of 4 mm….”

This sentence comes out of nowhere, the sentence before it talks about some deficiencies in physical properties of GIC. Why is the 4mm in the BCR suddenly relevant? This makes no sense.

RESPONSE 7: Transition from the description of the glass ionomer cement to the description of bulk-fill composites were provided as the sentence ‘Another alternative material to GIC used in pedodontics clinics are types of composites.’ was added.

POINT 8:  Finally: lines 50 to 54 establish that the GIC has problems, limitations such as “no conclusive evidence of caries inhibitory effect” or “it suspected tendency for microleakage” or “deficiencies in physical properties”… but then in lines 75-77 (only 20 lines later) they say that … “GIC has emerged as a prominent player through its SUPERIOR PHYSICAL, chemical and mechanical ATTRIBUTES, including fluorine release, adhesion to tooth tissue, biocompatibility, and its similarity to dentin tissue in terms of thermal expansion coefficient….”

So they think GIC are good, bad, or what? This is very confusing.

RESPONSE 8: Lines 50-54 describe the old version of glass ionomer cement. Lines 75-77 describe the new modified systems, the one that we used in our study (high viscosity glass ionomer cement). The word "superior" has been removed to clear the confusion.

POINT 9: In the last revision I also requested to add citations, they make specific statements without citations, two examples are:

Line 103: …”One of the drawbacks of composite materials polymerization, with 1-3% shrinkage during curing…”

Line 105: … Polymerization shrinkage causes post-restoration precision, failure of marginal adaptation and an increase in microleakage values…”

Response 9: As you mentioned in Point 6 that the introduction is very long, we omitted the sentences starting in (the old version of the manuscript) lines 103 and 105.

POINT 10: Line 182: what does the following sentence mean? “…The power analysis of this study was performed based on the two different materials…” What does this imply? What was the result of the analysis? what does it mean to write this?

Response 10: “The power analysis of this study was performed based on the evaluation periods to calculate the sample size. The sample size required for each group was determined to be at least 35 using G- power software version 3.1.9.7 for Windows (Heinrich Heine, Universität, Dusseldorf, Germany), for a power of 94%, the effect size of 1.73.” descriptive sentences added.

POINT 11: Writing “the first” and “the second” in lines 187 and 189 confuses when later they write “randomly” in line 190

Response 11:  Writing “the first” and “the second” changed to clear the confusion

POINT 12: Line 193. Double-blinded?  A double blind means that neither the participants nor the investigator, in this case, the dentist knows which intervention is receiving. This is not possible because the dentist used two protocols for the placement of the different materials and clearly knows what he is placing.

Response 12: A double-blind means that neither the participants (patients) nor the investigator (examiner) knows which intervention is receiving. Like the participants and examiners were unaware of the materials used in our study. (line 183, The examiners, who were not involved in the placement procedures, evaluated all the restorations at 6, 12, 18, and 24 months). However, the operators selected the treatments that were already blindly allocated to a group from an envelope and knew the material (treatment procedure) they were administering as you stated.

POINT 13: The first sentence of the results (line 254) says…” The ages of the children ranged from 10 to 12 years”… but this is not necessary because that is precisely a selection criterion “ line 154

Response 13:  changed to “The mean age of children was found to be 11.2 years”

POINT 14: In discussion, the same happens as with the introduction, it is necessary to reduce the discussion, everything that is said there must have a meaning and serve to discuss the results. In many cases, citations are missing as in the introduction.

Response 14: Discussion has been reduced and citations were checked

POINT 15: The first sentence of the discussion on lines 310-311 reads: …” Nowadays, EQUIA is a long-term clinical trials assessing GICs’ clinical performance in-vivo studies in permanent molar teeth are limited…”.

What does that mean?

Response 15:  EQUIA is one of the brand name of the GICs and this brand has been used in permanent teeth in vivo studies. This term is also used in many articles however to perform uniformity we have changed it to GICs.

POINT 16: What is the difference between the following sentences?

Lines 464-465: … the both GICs and  BRC composite restorative materials exhibited satisfactory results after 2 years of clinical service…..

Lines 473-474: … The GICs and BRC restorative materials display good clinical performance with a period of 24 months

Both are in the conclusion with only eight lines of separation

Response 16: “Lines 473-474: … The GICs and BRC restorative materials display good clinical performance with a period of 24 months” this sentence was deleted as the meanings are the same.

POINT 17: Finally, the following sentences are not a conclusion of the study; in any case, they could be part of the discussion:

475-476: …”Continued development of the mechanical and physical properties of the both restorative 475 materials will enable the construction of longer-lasting restorations…”

478-480: ”We believe that our study will provide useful guidance for future in-478 vivo and in-vitro studies on this subject and contribute to the further development of this 479 area in the science of pedodontics…”

Response 17: The mentioned sentences were moved to the discussion section.

POINT 18: I say it again, the work is very interesting, but the quality of the written document needs to be greatly improved.

Response 18: Requirements for improving the article and making it suitable for publication were made in line with the comments of the reviewers. Extensive rechecking of grammar, and punctuation was achieved with the help of professional English language editing services.

We sincerely appreciate the comments of the reviewers. According to your suggestion, the old manuscript was replaced with revised documents. We have tried to complete your comments, and we hope our replies to each comment are provided.

Round 3

Reviewer 4 Report

Thank you very much